# Comparison between Early Loaded Single Implants with Internal Conical Connection or Implants with Transmucosal Neck Design: A Non-Randomized Controlled Trial with 1-Year Clinical, Aesthetics, and Radiographic Evaluation

**DOI:** 10.3390/ma15020511

**Published:** 2022-01-10

**Authors:** Francesco Mattia Ceruso, Irene Ieria, Marco Tallarico, Silvio Mario Meloni, Aurea Immacolata Lumbau, Alessandro Mastroianni, Alessio Zotti, Marco Gargari

**Affiliations:** 1Department of Dentistry “Fra G.B. Orsenigo” Ospedale San Pietro F.B.F., 00189 Rome, Italy; f.m.ceruso@gmail.com (F.M.C.); alemastroianni11@gmail.com (A.M.); alessiozotti94@gmail.com (A.Z.); marco.gargari@gmail.com (M.G.); 2School of Dentistry, University of Sassari, 07021 Sassari, Italy; me@studiomarcotallarico.it; 3Surgical, Micro-Surgical and Medical Science Department, University of Sassari, 07021 Sardinia, Italy; melonisilviomario@yahoo.it (S.M.M.); alumbau@uniss.it (A.I.L.); 4Department of Clinical Science and Translational Medicine, University of Rome “Tor Vergata”, 00133 Rome, Italy

**Keywords:** dental implants, implant surfaces, oxidized, transmucosal, bone level

## Abstract

To evaluate the implant and prosthetic of two implants with different surfaces and neck design. Enrolled patients received bone level, 12° conical connection implants (Nobel Parallel, Nobel Biocare; NOBEL group) with anodized surface (TiUnite) and roughness of 1.35 μm, or transmucosal implant system (Prama, Sweden and Martina; PRAMA group) with convergent collar, ZIrTi surface, and roughness 1.4–1.7 μm. Both implants were made of pure grade IV titanium, with similar diameter and length, chosen according to the dentistry department availability and patient’s request. After early prosthesis delivery, patients were filled for at least one year. Outcome measures were: implant and prosthetic survival and success rates, physiological marginal bone remodeling, periodontal parameters and pink esthetic score (PES). Results: Fifteen patients were allocated and treated in each group. At the one-year follow-up, three patients dropped out, one in the NOBEL group and two in the PRAMA group. During the entire time of investigation, all implants survived and the prostheses were successful. No statistically significant differences were found in term of marginal bone loss, periodontal parameters, and aesthetics (*p* > 0.05). Conclusion: With the limitations of the present study, both implant systems showed successful clinical results. Finally, many other clinical and surgical variables may influenced marginal bone levels, implant survival, and periodontal parameters. More homogenous clinical trials with larger samples are needed to confirm these preliminary conclusions.

## 1. Introduction

As the request for immediate loading of dental implants has increased, companies are developing new dental implant surfaces with the purpose of shortening the time needed for osseointegration, while maintaining a high success rate. The concept of osseointegration was recently reassessed in view of the foreign body reaction theory. Following the latest research, osseointegration is a dynamic process between the implant surface and the healing capacity of the host [1]. Implant placement may activate a pattern of immune response named, type 2 inflammation, characterized by high levels of immunoglobulin E and eosinophils, and regulated by certain immune mediators [2]. Furthermore, regulatory adjustment of the immune system (or immunomodulation) strategies of macrophages in osseointegration is greater than expected [3,4]. Despite the high survival rates of titanium dental implants [5], it is necessary to further improve the implant-host relationship to maintain the foreign body equilibrium. The new goal for long-term successful implant therapy is to maintain unaltered the osseointegration process, in the form of unchanged peri-implant bone remodeling. In fact, the bone remodeling process plays a major role in evaluating implant success [6].

To achieve cell adhesion and speed up the osseointegration period, dental implant surfaces changed from smooth to rough. Several techniques that act to improve the roughness of the implant surface have been utilized, including but not limited to blasting, blasting plus acid-etching, and anodization [7]. Further, anodization may fast osseointegration because it incorporates calcium and phosphate ions on the implant surface [8,9].

Marginal tissues may respond to early or late loading with a significant crestal bone loss of about 1 to 2 mm in the first year, and then 0.2 mm yearly, respectively. Most of them depend on surgical, implant-related, and prosthetic factors [10]. This value was considered as clinically “normal” for the external hexagon implant-abutment interfaces, and it was also considered a measure of successful long-term implant treatment as conceived by Albrektsson and colleagues in 1986 observing the original Brånemark implants [11]. Later, the introduction of the conical internal connection of different degrees, in an attempt to overcome the drawbacks of original external-hexagon (EH) implants, showed excellent clinical results. This new interface design with built-in platform-switching, was demonstrate to reduce crystal bone loss [12,13]. Nevertheless, in view of the novel implant designs and more performant surfaces, the standard value should present a lower amounts of bone loss, both at the biological width establishment, and during function [14,15,16,17]. Recently, preliminary evaluation of a novel concept of connection between implant and abutment, featuring a vertical design, and named tissue development preparation technique, combined with screw-retained anchorage, have been proposed [18]. Despite its tissue-level design, with the implant-abutment interface placed coronally to the bone crest, the main differences compared to existing tissue-level implants are the convergent profile of the transgingival neck, inspired to the BOPT preparation, and the lack of a predetermined finishing line.

The aim of the present study was to compare marginal bone loss and survival rate by radiographic analysis and clinical parameters, around internal connection implants or intramucosal implant systems. The null hypothesis was that there were no differences in clinical and radiographic outcomes between the groups. The null hypothesis was tested against the hypothesis of difference. The present paper was written according to the STROBE guidelines.

## 2. Materials and Methods

This research was designed as a nonrandomized controlled trial, aimed to evaluate clinical and radiological data of partially edentulous patients, aged 18 years or older, able to sign an informed consent, in needed of at least one single implant-supported restoration to be placed in a healed site was asked to participate in this study. Principles stated in the Declaration of Helsinki of 2013 were respected. The present publication was approved by the ethics committee of Aldent University in Tirana (Protocol n°1/2021). All subjects had been informed about the protocol and signed informed consent. Enrolled patients, whose medical and sensitive data were anonymized, were treated at the Department of Dentistry “Fra G.B. Orsenigo” Ospedale San Pietro F.B.F., University of Rome “Tor Vergata”, Rome, Italy, from March 2018 to May 2019. Surgical and prosthetic procedures were performed by the same clinician (FMC). Preoperative cone beam computer tomography (CBCT) scan was obtained for all the potentially eligible patients to quantify bone volumes at the planned implant sites. Patients having sufficient bone volumes to receive a standard diameter implant, of 10 mm of length were considered eligible for the study. Postextractive sites must have been healing for at least three months before being treated. A complete list of exclusion criteria is reported in the Table 1.

Surgical and prosthetic procedures: One to two weeks prior to implant surgery, all the included patients underwent a professional oral hygiene therapy. Amoxicillin, or Clyndamicin in patients allergic to penicillin, were administered one hour before the surgical procedure. Immediately before implant placement, the patients rinsed with 0.2% chlorhexidine mouthwash for sixty seconds. After that, patients received local anesthesia (Articaine with adrenaline 1:100,000). Implant site was exposed with a crestal incision and a full-thickness flap elevation, than it was prepared according to the bone density and the manufacturer’s instructions. Patients were allocated to different groups using methods that are not random. Bone level, conical connection implants (Nobel Parallel, Nobel Biocare, Kloten, Swiss; NOBEL group) of 10 mm length and 4.3 mm diameter, or intramucosal implants of 10 mm length and 4.25 mm diameter (Prama, Sweden and Martina, Due Carrare (PD), Italy; PRAMA group) were placed according to dentistry department availability and patient’s request. Main implants characteristics are reported in Table 2. The bone quality was assessed by the surgeon during drilling and subjectively reported as hard, medium and soft.

All the implants were placed according to a one-stage protocol. In all the cases, healing abutments were immediately used (Figure 1).

Finally, the flap was closed with a resorbable 4.0 sutures around the healing abutment, and a periapical radiograph was taken. A soft diet was recommended for two weeks. Ibuprofen 400 mg were prescribed to be taken two to three times a day after meals. However, patients were instructed not to take them in absence of pain. In case of allergy, or of stomach diseases, one gram of paracetamol was prescribed instead. A 0.2% chlorhexidine mouthwash was prescribed to be use for one minute twice a day for 14 days. After 7 to 10 days, sutures were removed.

Eight weeks after implant placement, a screw-retained restoration was delivered. We checked the occlusal surface and kept it in light contact with the antagonist element. Periapical radiograph and clinical photos were taken (Figure 2).

Professional hygiene maintenance was delivered every six months after initial loading. Radiographs were taken yearly. Dental occlusion was evaluated at each follow-up visit (Figure 3 and Figure 4).

Patient allocation and statistical analysis: during the study period, two implant systems were available at the Department of Dentistry, “Fra G.B. Orsenigo” Ospedale San Pietro F.B.F., University of Rome “Tor Vergata”. Participants chose which group they wanted to be in. The researcher explained that both implants have similar characteristics. In case of no preference, participants were assigned to the groups by the researcher in a random draw, until there were two equal groups of 15 patients each.

Data were analyzed and compared with a pre-established analysis plan by a statistician with competence in dentistry. Comparisons between each time points were made by unpaired t-tests, to detect any changes in continuous outcomes. Dichotomous outcomes were compared using the Fisher exact test. All statistical comparisons were conducted at the 0.05 level of significance.

Primary outcome measures were the success rates implants and prostheses, and any biological and technical complications experienced during the entire follow-up period. Secondary outcomes measure were marginal bone loss, pink esthetic score (PES), and the periodontal parameters (bleeding index [BI] and plaque index [PI]). A brief explanation of all the outcome measures are reported in Table 3.

Implant and prosthetic failures, as well as, complications were assessed by the same clinical that performed all the procedures. Marginal bone levels and PES were evaluated, respectively by an independent radiologist and clinician, not previously involved in the study. Periodontal parameters were evaluated by an independent dental hygienist, at the hygiene maintenance visits.

## 3. Results

Thirty-three patients were screened for eligibility, but three patients refused to participate in the study. Thirty patients were finally enrolled and consecutively treated. According to the study design, two equally sized groups with 15 patients each, were obtained. However, one patient in the NOBEL group and two patients in the PRAMA group dropped-out after implants placement. Both patients refused to attend the planned visits due to the pandemic. A flow chart of the treated patient is reported in the Figure 5, and baseline characteristics of the treated patients are reported in the Table 4.

At the annual examination, all the implant-supported restorations were stable, without any complications. The implant and prosthetic survival and success rates were 100% for both groups.

At implant placement the mean marginal bone level was 0.04 ± 0.06 mm (0.00 to 0.07 mm) for the NOBEL group and 0.01 ± 0.02 mm (0.00 to 0.02 mm) for the PRAMA group. At the one-year follow-up examination, the mean marginal bone level was 0.99 ± 0.71 mm (0.61 to 1.36 mm) for the NOBEL group and 0.65 ± 0.48 mm (0.40 to 0.91 mm) for the PRAMA group. The mean marginal bone loss measured one year after loading was 0.99 ± 0.71 mm (0.61 to 1.36 mm) for the NOBEL group and 0.65 ± 0.48 mm (0.40 to 0.90 mm) for the PRAMA group. In both groups the differences between baseline was statistically significant (*p* < 0.05). Nevertheless, the difference in mean marginal bone loss between groups were not statistically significant (*p* = 0.192). Data are summarized in Table 5.

Pink esthetic score (PES), and the periodontal parameters (bleeding index [BI] and plaque index [PI]) were recorded in both groups at the one-year follow-up. No statistically significant differences were experience between groups in any of the measured outcomes. Data are reported in Table 6.

## 4. Discussion

The present prospective case series study was aimed to compare the clinical and radiographic data of conical connection implants featured with oxidized surface and conical connection, and intramucosal implant system with ZirTi surface, sand-blasted with zirconium oxide and etched with mineral acids, inserted in partially edentulous patients. One year after definitive prosthesis delivery, no statistically significant difference was experience between groups, in each of the tested outcomes. The null hypothesis that there are no differences between groups can be accepted.

The main limitation of the present study is inherent in the study design, namely, that two different implant systems were compared. Nevertheless, the authors were willing to compare implant systems available at the Department of Dentistry “Fra G.B. Orsenigo-Ospedale San Pietro F.B.F.”, University of Rome “Tor Vergata”. Other limitations are the small sample size and the short follow-up. This can be solved by continuing to enroll and follow the patients. The last limitation is that the data on vertical alveolar mucosal height have not been collected.

Despite these limitations, the results of the present research failed to find that one implant system was better than the other, meaning that correct treatment plan and accurate execution of all the surgical and prosthetic treatments are more important in the survival and success of the implant-supported rehabilitations. However, it should be considered that successful clinical and radiological results are found in both implant systems, and these results are in line with other reports presenting such data on these specific types of implant surfaces and interfaces [18].

The implant surface area can be modified by proper procedures, either by addition or subtraction [19]. The tested implants include the following characteristics. PRAMA implants are featured by ZirTi surface, sand-blasted with zirconium oxide and then etched with mineral acids. Through this treatment, the mean roughness of the implant surface is of about 1.4–1.7 μm. Furthermore, a recent histological study in humans showed an excellent mineralization directly onto the ZirTi surface, at 8 weeks, also in the woven bone [20]. On the other hands, the TiUnite surface (NOBEL group) presents a mean roughness of 1.35 μm [21]. This surface is obtained through a spark anodization in an electrolytic solution of phosphoric acid. The final result is a high crystallinity and phosphorus content (11% P) in its oxide layer, organized in a duplex oxide structure. The first layer is an inner barrier without pores, while the second layer is an outer porous layer with diameters and depths ranging between ≤4 microns and ≤10 microns [22,23].

Moderately rough implant surfaces are considered ideal to achieve osseointegration, due to a surface morphology that aims for high osteoconductivity and fast anchorage to the collagen matrix. Increase in dental implant surface roughness promotes the migration and retention of osteogenic cells, from the human body to the implant surface [24,25]. Inoue and co-workers have already showed that a surface roughness of 0.5 μm allows for fibroblast adhesion, while a greater roughness ranging from 0.5 to 1.5 μm, is necessary for osteoblast adhesion [26].

In the present study, two implant systems with different implant surface treatments were tested. Both sand-blasted and acid etched surfaces [27] as well as anodization surface [28] presented long term results, with at least 10 years of follow-up. History of periodontitis, combined with not fully adhering to the supportive periodontal therapy are the most important factor to influence the long-term results with sand-blasted and acid-etched implants [27]. Furthermore, anodized surface demonstrated successful results in case of immediate function [5,28,29,30].

In the present study, the tested implants also present different neck designs. The PRAMA implant has a convergent part characterized by a hyperbolic portion with a height of 2.00 mm and a cylindrical part of 0.80 mm, with no sharp edges [31]. Both portions featured microthreading, named an ultrathin threaded microsurface (UTM). The absence of sharp edges may allow the collagen fibers of the soft tissues to adhere to the titanium, preventing the accumulation of plaque around the junction with the abutment. Moreover, it facilitate the positioning of the prosthetic crown in any part of the transgingival section [32]. Controversially, NOBEL Biocare implants presents a 12-degree internal conical connection designed to accommodate a smaller diameter connection size, according to the palter switching concept [13,33]. NOBEL Biocare implants with internal conical connection are featured with in-build platform switching, and can be used with both cemented- and screw-retained implant-supported solutions. On the other hand, the transmucosal implant system can be used for both cemented or screw-retained single crowns with no restrictions in anterior sectors, while in posterior areas it is mandatory to close the prosthesis on the neck of the implant.

Several systematic reviews confirmed that two-piece bone level implants may suffer from microbiological colonization and inflammation at the implant–abutment interface [34,35]. Nevertheless, the results of the present study are in agreement with the conclusion of a recent systematic review by the Department of Stomatology of the University of Valenciano, which found no evidence of difference in marginal bone loss between transmucosal and bone-level dental implants [36]. Further studies are needed to evaluate the performance of the intramucosal implant system (PRAMA implant) compared with the conventional tissue-level design.

In the present study, although no differences were found in any of the tested outcomes, a mall trend of less marginal bone loss were found for the transmucosal implants. This results are in agreement with a retrospective study by Canullo and coauthors, where transmucosal implants demonstrated a suitable alternative to bone-level implants in the anterior area, up to 5 years after loading [37]. According to the aforementioned results, it is the authors’ opinion that, even if both implant systems presented successful clinical results, transmucosal implants may be suggested in the anterior area or in case of periodontally compromised patients, while bone level implants may be suggested in the middle, posterior area of healed patients, particularly in case of immediate loading. Accordingly, ten-year prospective and retrospective studies supported the use of bone level implants with TiUnite surface in well maintained patients [5,30].

## 5. Conclusions

With the limitations of the present study, the authors concluded that both implant systems showed successful clinical results. Finally, many other clinical and surgical variables may influenced marginal bone levels, implant survival, and periodontal parameters. More homogenous clinical trials with larger samples are needed to confirm these preliminary conclusions.

## Figures and Tables

**Figure 1 materials-15-00511-f001:**
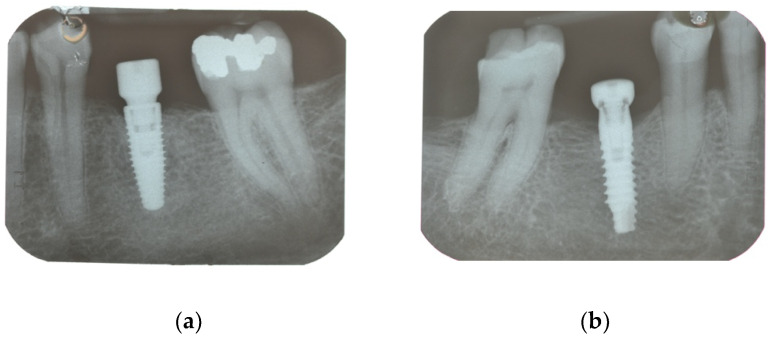
Radiographs of the implants with the screw healing at the time of insertion (**a**) Nobel Parallel; (**b**) Prama Sweden & Martina. Radiograph when the screw-retained restoration was delivered. Figures should be placed in the main text near to the first time they are cited. A caption on a single line should be centered.

**Figure 2 materials-15-00511-f002:**
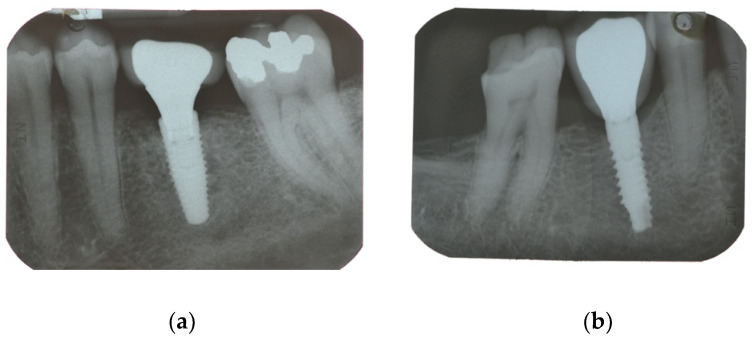
Radiographs when the screw-retained restoration was delivered (**a**) Nobel Parallel; (**b**) Prama Sweden & Martina.

**Figure 3 materials-15-00511-f003:**
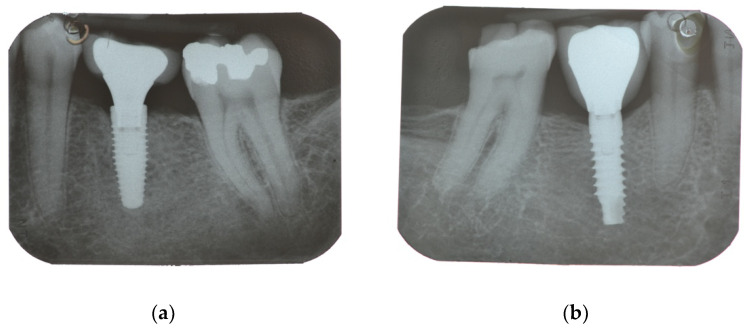
Radiographs after one year. (**a**) Nobel Parallel; (**b**) Prama Sweden & Martina.

**Figure 4 materials-15-00511-f004:**
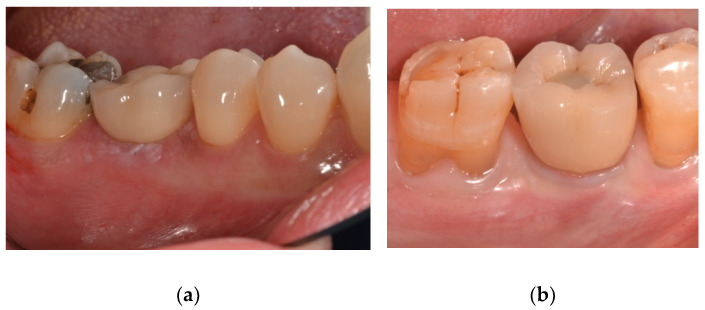
Photos when the screw restoration was delivered. (**a**) Nobel Parallel; (**b**) Prama Sweden & Martina.

**Figure 5 materials-15-00511-f005:**
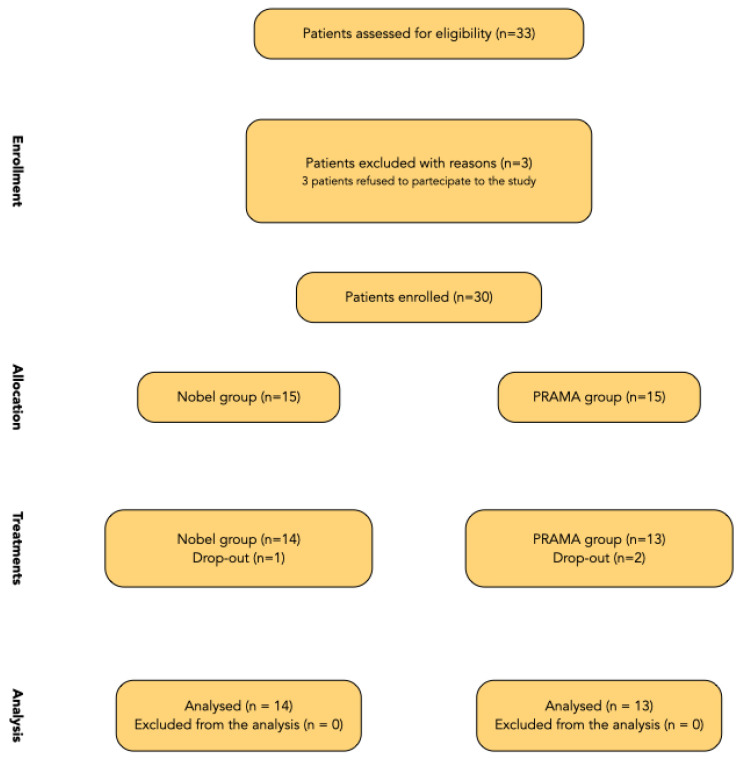
Flow chart.

**Table 1 materials-15-00511-t001:** Exclusion criteria.

General contraindications to implant surgery (e.g., uncontrolled diabetes)
Patients irradiated in the head and neck area
Immunosuppresed or immunocompromised patients
Patients treated or under treatment with intravenous amino-biphosphonates
Patients with untreated periodontitis
Immediate, postextractive implants
Patients with poor oral hygiene and motivation
Pregnancy or nursing
Substance abuser, psychiatric problems or unrealistic expectations
Patients unable to be followed-up
Patients with infection in the area intended for implant placement

**Table 2 materials-15-00511-t002:** Main implant characteristics.

NOBEL Biocare Parallel	PRAMA
Pure grade IV titanium	Pure grade IV titanium
TiUnite anodized surface	Pure ZIrTi Surface
12° of conical connection	Transmucosal with a convergent part of 2 mm and a cylindrical part of 0.8 mm, without sharp edges
Roughness 1.35 μm	Roughness 1.4–1.7 μm

**Table 3 materials-15-00511-t003:** Outcome measurements.

Primary Outcome	Implant failure	An implant was considered a failure if it presented any mobility, tested by tapping or rocking the implant head with a hand instrument and/ or any signs of radiolucency and/or fracture on an intraoral radiograph taken with a paralleling technique strictly perpendicular to the implant bone interface. The implant stability were assessed at initial loading and yearly without the prostheses removed.
	Prosthetic failure	A prosthesis is considered a failure if it needs to be replaced by an alternative prosthesis.
	Complication	Any biological (pain, swelling, suppuration, etc.) and/or mechanical complication (fracture of the framework and/or the veneering material, screw loosening, etc.) was considered.
Secondary Outcome	Marginal bone loss (MBL)	Marginal bone level changes were assessed using intraoral digital periapical radiographs at implant placement (baseline), and at after one year on function. Intraoral radiographs were taken with the parallel technique with customized holder. All the radiographs were evaluated under routine conditions. The software has been calibrated for every single image using the known distance of the implant diameter or length. The distance from the reference point at the implant neck to the first bone to implant contact were taken as the horizontal marginal bone level at both mesial and distal aspects. The average radiographic values of mesial and distal measurements were taken for each implant. Variation of the marginal bone levels at different time was taken as marginal bone loss.
	BI and PI	Soft tissue parameters (BI and PI) around the implant/abutment interfaces were assessed yearly using a plastic periodontal probe (Plast-o-Probe, Dentsply Maillefer, Ballaigues, Switzerland). The BI were evaluated at four sites around each implant (mesial, distal, buccal and lingual) according to the Mombelli Index. The bleeding elicited 20 s after the careful insertion of a periodontal probe 1 mm into the mucosal sulcus, parallel to where the abutment wall will be assessed (0 = no bleeding; 1 = spot bleeding, 2 = linear bleeding, and 3 = spontaneous bleeding). The PI, defined as the presence of plaque (yes/no) on the abutment/restoration complex, was measured by running the periodontal probe parallel to the abutment surfaces, and scored at one site for implant. An independent blinded dental hygienist performed all periodontal measurements.
	PES	The aesthetic evaluation was performed according to the pink esthetic score (PES) on the vestibular and occlusal pictures taken including at least one adjacent tooth per side. The values will be assessed annually after definitive loading. Seven variabilities (mesial papilla, distal papilla, soft tissue level, soft tissue contour, alveolar process deficiency, soft tissue color and texture) were assessed at 0 to 2 score (0 being poorest and 2 being the best) by a blind outcome assessor.

**Table 4 materials-15-00511-t004:** Baseline characteristics between groups.

	NOBEL Parallel	PRAMA	*p* Value
Mean age (SD)	49.4(10.6)	47.5(13.4)	0.6759
Sex (Male/Female)	6/9	8/7	0.7152
Smokers	2	1	1.0
Drop-Out	1	2	1.0
Maxilla/Mandible	9/6	7/9	0.4795
Molar implants	9	7	0.4795
Premolar implants	5	7	0.7160
Anterior implants	1	1	1.0

**Table 5 materials-15-00511-t005:** Mean marginal bone levels between groups.

	NOBEL Parallel	PRAMA	*p* Value
Implant placement	0.04 ± 0.06 (0.00 to 0.07); *n* = 15	0.01 ± 0.02 (0.00 to 0.02); *n* = 15	0.128
One-year follow-up	0.99 ± 0.71 (0.61 to 1.36); *n* = 14	0.65 ± 0.48 (0.40 to 0.91); *n* = 13	0.166
Difference	0.96 ± 0.72 (0.58 to 1.34)	0.65 ± 0.48 (0.40 to 0.90)	0.192

**Table 6 materials-15-00511-t006:** Comparison of BI, PI and PES between groups.

	NOBEL Parallel (*n* = 14)	PRAMA (*n* = 13)	*p* Value
BI	3	3	1.0
PI	4	3	1.0
PES	9.79 ± 2.61 (8.42 to 11.15)	10.46 ± 2.30 (9.26 to 11.66)	0.481
*p* Value	0.000	0.000	

## Data Availability

All data are available upon request of the author.

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
