# Peer review of "Comparison between Early Loaded Single Implants with Internal Conical Connection or Implants with Transmucosal Neck Design: A Non-Randomized Controlled Trial with 1-Year Clinical, Aesthetics, and Radiographic Evaluation"

_materials, 2022, doi:10.3390/ma15020511_

Round 1
Reviewer 1 Report
The study was evaluated marginal bone loss and survival rate of two different implant brand. The study was interesting but major corrections were required.
Transmucosal neck and difference from tissue level implant should have more explaine.
The hypothesis and aim should emphasized
Flow chart must be added to manuscript
Material method should be divided the subcategory such as eligibility criteria, clinical and radiographic measurements, surgical process statictical analyzes etc.
Table 4 and table 5 accidentally wrong titled, I guess. And Tables should be more explained such as baseline and first year follow up.
In which regions were the implants placed?
Have data on vertical alveolar mucosal height been collected? If not, this should also be presented in the discussion as a limitation.
Information about the demographic data of the implants should be given - was it a single dental implant or was it also used as a bridge support - was it completely placed in the posterior region.
Demographic data of the included patients should be presented. Why did bone level and tissue level implants is compare?
As far as I understand, PRAMA implants offer a new concept, but would it be more accurate to compare them with a different implant (Straumann tissue level or Nobel Biocare On1 Abutment concept) This issue should also be discussed in the conclusion.

Author Response
The study was evaluated marginal bone loss and survival rate of two different implant brand. The study was interesting but major corrections were required.
Transmucosal neck and difference from tissue level implant should have more explaine.
Thanks for the good observation.
In the introduction section has been added:
"Despite its tissue-level design, with implant-abutment interface placed coronally to the bone crest, the main differences compared to existing tissue-level implants are the convergent profile of the transgingival neck, inspired to the BOPT preparation, and the lack of a predetermined finishing line."
In the discussion section has been added:
"Several systematic reviews confirmed that two-piece bone level implants may suffer of microbiological colonization and inflammation at the implant-abutment interface [34, 35]. Nevertheless, the results of the present study are in agreement with the conclusion of a recent systematic review by the Department of Stomatology of the University of Valenciano, that found no evidence of difference in marginal bone loss between transmucosal and bone-level dental implants [36]. Further studies are needed to evaluate the performance of the the intramucosal implant system (PRAMA implant) compared with the conventional tissue level design."
The hypothesis and aim should emphasized
Thanks. The hypothesis has been emphasized. Introduction: "The null hypothesis was that there were no differences in clinical and radiographic outcomes between the groups. The null hypothesis was tested against the hypothesis of difference." Discussion: "The null hypothesis that there are no differences between groups can be accepted."
Flow chart must be added to manuscript
A flow chart of the treated patient is reported in the figure 5.
Material method should be divided the subcategory such as eligibility criteria, clinical and radiographic measurements, surgical process statictical analyzes etc.
Thanks a lot. Nevertheless, the manuscript has been written according to the CONSORT guidelines, so all the paragraphs are well organized. I regret to the editorial office if they prefer subcategories.
Table 4 and table 5 accidentally wrong titled, I guess. And Tables should be more explained such as baseline and first year follow up.
Thanks a lot. Tables 4 and 5 have been inverted. MBL has been reported at different timepoints. Table 4 just report outcomes at 1 year because, BI, PI, and PES were not evaluated at baseline.
In which regions were the implants placed?
Baseline characteristics have been added in the table 4.
Have data on vertical alveolar mucosal height been collected? If not, this should also be presented in the discussion as a limitation.
Unfortunately, data on vertical tissue height have not been collected. It has been added as limitation.
Information about the demographic data of the implants should be given - was it a single dental implant or was it also used as a bridge support - was it completely placed in the posterior region.
All of the implants received single crowns. This has been added in the title and materials and methods. Moreover, a baseline characteristics have been added in the table 4.
Demographic data of the included patients should be presented. Why did bone level and tissue level implants is compare?
Thanks. Baseline characteristics have been added in the table 4.
As far as I understand, PRAMA implants offer a new concept, but would it be more accurate to compare them with a different implant (Straumann tissue level or Nobel Biocare On1 Abutment concept) This issue should also be discussed in the conclusion.
"Several systematic reviews confirmed that two-piece bone level implants may suffer of microbiological colonization and inflammation at the implant-abutment interface [34, 35]. Nevertheless, the results of the present study are in agreement with the conclusion of a recent systematic review by the Department of Stomatology of the University of Valenciano, that found no evidence of difference in marginal bone loss between transmucosal and bone-level dental implants [36]. Further studies are needed to evaluate the performance of the the intramucosal implant system (PRAMA implant) compared with the conventional tissue level design."
Reviewer 2 Report
The revised work has an introduction in which the performance of the work is justified. The methodology used is correct and meets the requirements of patient research. The conclusions respond to the stated objectives. The only change I suggest is presenting the tables on a single sheet or dividing them into several tables. It would greatly facilitate the reading and understanding of them. It is also interesting to add a flow chart in the first section of the results. From my point of view, the work only needs these minor revisions.Author Response
The revised work has an introduction in which the performance of the work is justified. The methodology used is correct and meets the requirements of patient research. The conclusions respond to the stated objectives. The only change I suggest is presenting the tables on a single sheet or dividing them into several tables. It would greatly facilitate the reading and understanding of them. It is also interesting to add a flow chart in the first section of the results. From my point of view, the work only needs these minor revisions.
Thanks a lot. I regret to the editorial office for this suggestion. From our point off view, we strictly follows the journal guidelines. Thanks.
Reviewer 3 Report
This is an interesting study. The authors have addressed their limitations adequately. The duration of the study is very short.
The examiner/ examiners, reproducibility/standardization has not been mentioned.
For repeated measures of the radiographic measurements or clinical parameters, instead of mean scores, site to site comparison would be more valid and reliable.
Author Response
This is an interesting study. The authors have addressed their limitations adequately. The duration of the study is very short.
This is reported as limitation.
The examiner/ examiners, reproducibility/standardization has not been mentioned.
For repeated measures of the radiographic measurements or clinical parameters, instead of mean scores, site to site comparison would be more valid and reliable.
Honestly I did not understand what you mean. Radiographic measure are continuous outcomes and according to the protocol have been tested using unpaired test. Dichotomous outcomes have been tested using Fisher exact test.
Round 2
Reviewer 1 Report
.